Environmental conditions influencing the abundance of the salmonid ectoparasite Salmincola californiensis across upper Willamette River Reservoirs, Oregon

Antonelli Kelsi 1
Murphy Christina christina.murphy@oregonstate.edu 2
Pollock Amanda 1
Arismendi Ivan 1
1 Department of Fisheries, Wildlife, and Conservation Sciences, Oregon State University , Corvallis , OR , United States of America
2 Maine Cooperative Fish and Wildlife Research Unit, U.S. Geological Survey , Orono , ME , United States of America
Esteban María Ángeles
Electronic publication date: 2025 Apr 7
Publication date: 2025
Volume: 13
Electronic Location ID: e19228
Received 2024 Dec 5; Accepted 2025 Mar 7
Copyright: ©2025 Antonelli et al.
Copyright year: 2025
Copyright holder: Antonelli et al.
License: This is an open access article, free of all copyright, made available under the Creative Commons Public Domain Dedication. This work may be freely reproduced, distributed, transmitted, modified, built upon, or otherwise used by anyone for any lawful purpose.
License URL: https://creativecommons.org/publicdomain/zero/1.0/

Keywords: Chinook Salmon, Light trap, Impoundment, Gill maggots, Copepodid, Dam operations, Stratification

Funding: US Army Corps of Engineers and an Oregon State University Graduate Assistantship This work was supported by the US Army Corps of Engineers and an Oregon State University Graduate Assistantship. The funders had no role in study design, data collection and analysis, decision to publish, or preparation of the manuscript.

==============================
The freshwater copepod Salmincola californiensis is an ectoparasite of Pacific salmon and trout (Oncorhynchus spp.). High levels of infection by this parasite can significantly damage gills and result in blood loss, affecting the fitness and survival of hosts, and it may hinder recovery efforts of threatened and endangered salmonids. Juvenile salmonids that rear in reservoirs have been reported to experience higher rates of infection than their stream-dwelling counterparts. To date, the relationship between reservoir environmental conditions and infection rates remains poorly understood. Here, we evaluated sampling methods designed to capture S. californiensis copepodids, the free-swimming infectious life stage of this parasitic copepod, and develop predictive models of parasite abundance in reservoirs. We used light traps to collect 675 zooplankton samples from various sites and depths in Cougar, Lookout Point, and Fall Creek reservoirs, Oregon, USA during five months in 2020. We tested several predictive models of parasite abundance using environmental covariates obtained concurrently during plankton surveys (e.g., temperature, ambient light and water clarity, reservoir plankton profiles, and water flow). Our models showed a strong influence of water temperature on abundance of copepodids, with maximum initial abundance occurring at the mean water temperature 15.2 °C. At that temperature, August abundance was estimated to be 58.6 copepodids per site in Cougar Reservoir, 43.8 copepodids per site in Fall Creek Reservoir, and 3.5 copepodids per site in Lookout Point Reservoir. Water temperature and reservoir outflow both affected population growth of copepodids: increased water temperature was associated with an increase in population growth rate while an increased reservoir outflow was associated with a decrease in population growth rate. The efficacy of our sampling method was influenced by percent of moon fullness, whether the sample site was above or below the thermocline, water temperature, and number of juvenile sculpin fishes (Cottus spp.) captured in the trap. The mean probability of detecting a single copepodid if one was present was 0.042, but detection probability increased to 0.29 under ideal trap set conditions of warmer water, location above the thermocline, and a full moon. Biologists and managers could use these methods to assess the presence and abundance of S. californiensis in other locations, or to inform reservoir operations to reduce potential salmonid infections.

Introduction

While the word “parasite” often has negative connotations, most parasites evolved alongside their hosts and are a natural part of a healthy ecosystem (Combes, 1996). For example, a parasite can stabilize a food web by preferentially attacking the more abundant of two predator species, thus allowing those two predator species to co-exist (Combes, 1996). Although they can play critical roles in food webs, parasites are often cryptic, and therefore understudied (Scholz & Choudhury, 2014). However, interest in parasites can increase when an environmental change occurs, whether climatic or human-induced, and the balance of an ecosystem is altered (Lepak et al., 2021). Under these disturbed conditions, the impact of a previously unnoticed parasite can change, and observations may indicate a dramatic increase in the effect of the parasite on its host (Holmes, 1996). The alteration of stream habitat and creation of reservoirs in the Pacific Northwest are arguably one such example, as the affected aquatic ecosystems may be shifting interactions between the endemic parasite Salmincola californiensis and one of its host species, Chinook Salmon (Oncorhynchus tshawytscha).

S. californiensis is a freshwater parasitic copepod that exclusively attaches to salmonids of the genus Oncorhynchus, typically on fins, gill filaments, and interior surfaces of branchial cavities (Kabata, 1969). The parasitic life cycle of a female (Fig. 1), once attached, is completed on a single fish host (Kabata & Cousens, 1973). Males reach less than half the size of females and attach with the attachment filament until their final molt, whereupon they use their second maxillae and the maxillipeds to grasp the host and seek a female to fertilize (Fasten, 1921; Kabata & Cousens, 1973). The parasite spreads primarily via lateral transmission, though there is still debate over the potential for autoinfection (Neal et al., 2021). The copepodid stage is the free-swimming infectious stage that seeks a host; copepodids must find a suitable host to survive, with the time to mortality linearly dependent on water temperature (Murphy, Gerth & Arismendi, 2020). While mild infections may be unnoticed (Murphy et al., 2020a), heavy infections on or near the gills can cause tissue destruction and likely decrease gill function (Herron, Kent & Schreck, 2018). This may result in anemia, decreased resilience, or mortality of the fish host, especially when other simultaneous stressors exist, such as for smolts transitioning to saltwater (Pawaputanon, 1980; Sutherland & Wittrock, 1985; Neal et al., 2021).

Figure 1 Salmincola californiensis life-cycle illustrated by K. Antonelli based on Kabata & Cousens (1973), updated to include pre-host attachment stages including the recently documented nauplius stage.

The free-swimming copepodid is the targeted life stage for light traps in this manuscript.

Though the native range of S. californiensis is not well documented, it is assumed that it corresponds to the native distribution of its hosts. S. californiensis has been reported throughout the Pacific Northwest, Midwestern watersheds, and the Great Lakes in North America (Bartos, 2021). Reservoir habitats have been associated with increasing prevalence and intensity of S. californiensis infections in Colorado and Oregon (Hargis et al., 2014; Monzyk, Friesen & Romer, 2015). For example, in reservoirs of the Willamette River Basin, Oregon, adult Chinook Salmon are collected at the base of dams as they travel upriver to spawn, then transported above the dam and released into historical spawning habitats. The progeny of these previously stream-type salmonids can enter downstream reservoirs as small subyearlings (<40 mm fork length), where they may feed and grow for several months before passing below dams during regular seasonal draw-down in fall (Keefer et al., 2012; Romer et al., 2012; Murphy et al., 2019). Chinook Salmon that spend time growing in Willamette River Basin reservoirs appear to experience higher infection prevalence (percentage of hosts infected) and intensity (number of parasites per infected host) than both their stream-dwelling counterparts (Monzyk, Friesen & Romer, 2015) and average historical infection prevalences and intensities (Murphy et al., 2020a). Increased infection can lead to decreased fitness and survival of juvenile and smolts (Pawaputanon, 1980; Herron, Kent & Schreck, 2018; Neal et al., 2021), hampering conservation and recovery efforts of Endangered Species Act (ESA)-listed threatened Chinook Salmon populations from the upper Willamette River (Monzyk, Friesen & Romer, 2015; NOAA Fisheries, 2021).

Although research has documented the impacts of these parasites to fish, very little is known about the abundance of S. californiensis within reservoirs. The increased infection prevalence and intensity in reservoir environments is hypothesized to occur for a variety of reasons. Reservoirs have lower flow velocities compared to streams. Without strong, flushing flows, it is likely easier for copepodids to remain attached in the branchial cavity (McGladdery & Johnston, 1988). Low flow velocities may also allow for efficient reinfection of the host by multiple generations of copepods (McGladdery & Johnston, 1988). In addition, Chinook Salmon schooling at the forebay of reservoirs could increase the chance of lateral transmission (Beeman, Hansen & Sprando, 2015; Monzyk, Friesen & Romer, 2015). Host density was associated with increased Salmincola spp. prevalence in recent stream studies along with greater flows, although correlations between host fish distributions and stream velocity may explain this unexpected relationship (Hasegawa & Koizumi, 2021). Reservoirs, being much deeper than streams, also add a larger vertical component to potential interactions between fish and copepodids. Diel vertical migrations of Chinook Salmon (14–23 m; Murphy et al., 2020b) may cross through depths that contain high concentrations of S. californiensis copepodids (Monzyk, Friesen & Romer, 2015; Murphy et al., 2019). However, the precise reservoir conditions associated with greater S. californiensis abundance remain unknown.

Here, we use seasonal sampling at multiple depths in three reservoirs located in the upper Willamette River Basin to predict the abundance of this small, elusive, free-swimming copepodid life stage of S. californiensis. We use a model selection approach that includes covariates (concurrently measured) that account for environmental conditions potentially affecting the abundance of S. californiensis. Previous laboratory experiments indicated a relationship between copepodid occurrence and water temperatures near 16.5 °C (Murphy et al., 2023), so we expect to see confirmation of this relationship in natural environments. We also hypothesize the effect of various covariates on detection probability. S. californiensis copepodids exhibit positive phototaxis and have been captured during previous field testing of LED light traps (Murphy et al., 2022), so we anticipate high amounts of ambient light from the surface during the day or a bright moon at night may affect trapping efficiency by making the LED lights less attractive by comparison. We also expect the numbers of Leptodora spp., a genus of predatory zooplankton common in these reservoirs, and juvenile sculpin (Cottus spp.) caught in the light traps to decrease detection probability, as they might consume captured copepodids in these traps before sample collection and preservation.

Our work is intended to inform management strategies whose goal is to identify and ultimately mitigate negative effects of this parasite on threatened Chinook Salmon, as current management options are limited. For example, the removal of these dams may not be feasible, as these reservoirs are necessary for flood control and agriculture in the region (US Army Corps of Engineers, 2015). Instead, this research could provide insights toward exploring indirect reservoir water management strategies, or direct capture of copepodids, without handling threatened fishes, as potential tools to minimize infection prevalence, infection intensity, and resulting gill damage to Chinook Salmon by S. californiensis.

Materials & Methods

Site description

Our sampling occurred in three oligo-mesotrophic, low-to-moderate productivity, reservoirs from the upper Willamette River Basin including Cougar Reservoir, Lookout Point Reservoir, and Fall Creek Reservoir, western Oregon, USA (Fig. 2). All three reservoirs are owned and operated by the US Army Corps of Engineers (USACE) and are the result of dams built primarily for reducing flood risk, but they are managed for multiple uses. In the spring, reservoirs conventionally experience water rises for flood risk management, recreation, and water storage to support downstream water releases during the dry summer months. In fall, reservoirs are conventionally drawn down in preparation for heavy rains and potential flooding during fall and winter months (US Army Corps of Engineers, 2022a; US Army Corps of Engineers, 2022b; US Army Corps of Engineers, 2022c). All three reservoirs support native salmonid populations, including Chinook Salmon, Rainbow Trout (Oncorhynchus mykiss), and Cutthroat Trout (Oncorhynchus clarkii) (Monzyk, Friesen & Romer, 2015).

Figure 2 Placement of light trap lines and vertical distribution of light traps.

(A) Trap line locations (insets; orange circles) near the dams (brown rectangles) in Fall Creek, Cougar, and Lookout Point reservoirs. Trap line locations in Fall Creek and Lookout Point reservoirs appear clustered because the surface area of both reservoirs at the time of sampling was smaller due to annual drawdown than is pictured in these base maps. (B) Diagram of the light trap line located at each orange point in the reservoir. Trap depths were initially centered around 16.5 °C until the annual breakdown of the thermocline. After this they were set from the surface. Traps were spaced five m apart and held vertical by weighted belts. Light would shine through the glass funnel at the bottom of the trap.

Cougar, Lookout Point, and Fall Creek reservoirs have been operational for several decades. Cougar Reservoir is an impoundment of the South Fork McKenzie River near the town of Blue River, Oregon that was completed in 1963. This water body averages 52 m in depth and has a 4.57 km2 surface area when full (Johnson, 1985). The dam contains a temperature control tower, which can selectively draw water from different levels of the reservoir for release downstream (US Army Corps of Engineers, 2022a). Lookout Point Reservoir impounds the Middle Fork Willamette River, downstream of the Willamette Fish Hatchery and directly upstream of Dexter Reservoir, near the town of Lowell, Oregon. The dam was completed in 1954, and the reservoir averages 32 m deep with a surface area of 16.53 km2 when full (US Army Corps of Engineers, 2022c). Lastly, Fall Creek Reservoir, also near the town of Lowell and just north of Lookout Point Reservoir, was completed by impounding Fall Creek in 1965 and averages 20 m in depth with a surface area of 6.94 km2 (US Army Corps of Engineers, 2022b). It is primarily fed by Fall Creek and Winberry Creek (Johnson, 1985). Unlike the other two reservoirs, management beginning in 2011 has included draining the reservoir to streambed in late fall with the intention of improving downstream passage of outmigrating juvenile salmon and reducing predatory non-native warmwater fishes in the reservoir (Murphy et al., 2019).

Sample design and collection

To meet the requirements of the N-mixture abundance modeling approach (see below; Royle, 2004), our sampling design followed three rules. First, sites were selected to approximate randomization and were assumed to be independent of one another. Second, each site was surveyed multiple times per season. Third, season was defined as a period when the population was closed (state of the site, occupied or not occupied by S. californiensis, did not change).

We collected zooplankton samples using light traps (Murphy et al., 2022). Each trap was composed of a blue light-emitting diode (LED) light suspended within a short length of polyvinyl chloride (PVC) pipe and enclosed with a dark rubber lid on one end and an inverted, clear glass funnel on the other (Fig. S1). Zooplankton were attracted to the blue light and the smell of fish fins/opercles (added to specifically attract S. californiensis and minimize bycatch), guided to the center of the funnel by its inverted orientation, and allowed to enter via the hole at the funnel’s center. The light traps retained zooplankton with a funnel escape hole too small to locate once inside the trap. Traps were assembled in water baths using filtered water from each respective site, which created a vacuum seal prior to each set; this seal remained until after the trap was retrieved from the reservoir and the rubber lid was removed. As fish may be captured using light traps, trapping was conducted under Oregon State University IACUC # 5163 and NOAA and ODFW permit #22982.

We set traps in each reservoir monthly from August to December 2020, except when reservoir access was not possible due to wildfires or reservoir drawdown (Table 1). We added trap lines to the accessible reservoirs when we were unable to sample all three in a given month. Ideally, sampling would have taken place as soon as the reservoir began to stratify in spring, but logistical complications due to the COVID-19 pandemic delayed sampling until August. Each month, traps were set, left to attract zooplankton for 48 h, and then retrieved. Their contents were poured through a 106-µm sieve, rinsed from the sieve into 250 ml bottles with a wash bottle and funnel, and preserved with 95% ethanol. Traps were immediately reset, and the process was repeated, for a total of two “passes” per month at each sampled site. Both sampling events were completed within 96 h for each month. This short time interval allowed us to assume the S. californiensis population was closed during the sampling period.

Table 1 Total Salmincola copepodids captured in light traps by month and reservoir.

Sampling did not occur for Cougar Reservoir in September due to wildfire or Fall Creek Reservoir in October–December due to annual fall drawdown for fish passage.

Month	Reservoir	Total copepodids captured	
August	Cougar	32	
Fall Creek	8	
Lookout Point	7	
September	Cougar	Not sampled	
Fall Creek	112	
Lookout Point	3	
October	Cougar	139	
Fall Creek	Not sampled	
Lookout Point	6	
November	Cougar	71	
Fall Creek	Not sampled	
Lookout Point	0	
December	Cougar	41	
Fall Creek	Not sampled	
Lookout Point	1	

Near the forebay of each reservoir, we set traps on five to eight trap lines, spaced apart with a minimum distance of 30 m between lines so they would be independent of one another (Fig. 2A). Between one and five traps (quantity dependent on depth and temperature gradient of reservoir) were attached five meters apart on a vertical trap line that was suspended by a buoy and held in place by a concrete anchor (Fig. 2B). Initial testing indicated that light from the traps did not extend beyond 2.5 m below the trap, so we assumed independence with five m spacing between them. We centered trap depths around 16.5 °C until annual thermocline breakdown, due to high concentrations of S. californiensis being captured at this temperature in previous test sampling under laboratory conditions (Murphy et al., 2023).

We also collected concurrent environmental and ecological data for each reservoir immediately prior to setting the first of the two trap sets each month. These covariates included reservoir depth profiles for temperature and light. We took light measurements using a light meter probe (Licor LI-192; Lincoln, NE) lowered at one m depth intervals. Light measurements were used to calculate the average light extinction coefficient and create light profiles for monthly samplings at each reservoir. We recorded temperature by slowly lowering a continuous multiprobe sonde (YSI EXO1; Yellow Springs, OH) through the water column, sampling at 2 s intervals which corresponded to approximately 0.5 m resolution. We fit polynomial lines to the temperature data using R statistical software version 1.4.1103 (R Core Team, 2020) to construct temperature profiles and then calculated the depth at which temperature changed the fastest to determine thermocline position. We recorded light trap characteristics, such as funnel size, trap type, and condition of trap after retrieval. We did not include water chemistry measurements (pH, dissolved oxygen, etc.) in our covariate suite, as water chemistry was nearly uniform horizontally and vertically in these oligotrophic lakes over the depths we set traps (0–28 m). The percentage of moon fullness at midnight following trap setting was estimated using an online moon phase calculator (https://www.mooncalc.org/). Reservoir outflow data were obtained from the USACE website (https://www.nwd-wc.usace.army.mil/dd/common/dataquery/www/) in cubic feet per second (ft3/s), which we then converted to cubic meters per second (m3/s) for analysis.

In the laboratory, we filtered samples using stacked sieves with a 500 µm mesh size on top to filter out larger organisms, such as Daphnia and mature calanoid copepods, and a 106 µm mesh size on the bottom to retain Salmincola copepodids while removing rotifers and other small phytoplankton. We then searched these mid-sized filtrates in their entirety under dissecting microscopes (20x magnification) for S. californiensis copepodids and recorded copepodid counts for each sample. These data were used as the response variable in the abundance models (see section below). We then recombined the filtered parts, split the samples into smaller subsamples, and identified and counted all other non-target zooplankton taxa (Antonelli et al., 2024). Non-target zooplankton data were used as covariates in models.

While other species of copepods, such as Diacyclops thomasi and Leptodiaptomus spp., exist in these reservoirs and were captured in our samples, they are non-parasitic. The morphology of these copepods is distinct from that of S. californiensis. Salmincola edwardsii is a closely related parasitic copepod, but it affects only fish of the Salvelinus genus, and S. edwardsii adults have not been detected in the Willamette River Basin (Murphy et al., 2020a). While we cannot fully rule out the existence of S. edwardsii on upstream Salvelinus species, it is extremely unlikely that any potential upstream S. edwardsii would make a substantial contribution to overall copepodid abundance in the downstream end of these reservoirs. Based on this information, we assumed that all Salmincola copepodids captured in our samples were S. californiensis.

Modeling approach and implementation

We used copepodid count data paired with environmental data to model the abundance and detection of S. californiensis. As with other ecological surveys of cryptic or small animals, even the best sampling methods may not always detect an organism when it is present. The free-swimming stage of S. californiensis exhibits rapid movements (Murphy & Arismendi, 2023) and is relatively rare when compared to other zooplankton, resulting in a lack of detection using traditional tow net zooplankton sampling. Even in light traps, it is rare relative to other zooplankton taxa. Unless specifically accounted for, this imperfect detection can lead to underestimates of abundance and incorrect inferences about relationships between animal presence and environmental covariates.

We used N-mixture models (Royle, 2004; Dail & Madsen, 2011) to estimate S. californiensis abundance and detection probabilities. This model type was developed to estimate the species abundance of unmarked animals among sample sites, while investigating potential covariates influencing that abundance. N-mixture models also allow our analysis to encompass multiple “seasons” (Dail & Madsen, 2011). A multi-season framework allowed us to account for imperfect detection and estimate abundance of this infrequently detected species. N-mixture modeling has been used successfully to model the abundance of other hard-to-spot animals (Rozylowicz et al., 2024; Madsen & Royle, 2023).

Assumptions of N-mixture models include population closure, individuals only being counted once during a survey, homogenous detection of individuals, and no unaccounted-for heterogeneity in abundance or detection (Royle, 2004). We addressed the first two assumptions by setting light traps within a short time frame (96 h) each month so the population could be assumed closed, and by removing S. californiensis copepodids from samples after identification to prevent double counting. It is unlikely that individual copepodids would have different detection probabilities since at this infectious stage all individuals are motivated to find a host. We accounted for abundance and detection heterogeneity by testing the inclusion of multiple environmental and biological covariates.

These N-mixture models consist of three distinct submodels, each using one of the following response variables: (1) initial abundance, (2) population growth, and (3) detection probability. We used covariates in each submodel, but did not include highly correlated covariates (r > 0.6) within the same submodel to avoid multicollinearity. Covariates included three levels of data: site-level (variables that change only between sites—light trap locations), season-level (variables that change only between months), and observation-level (variables that change only between our two trapping events each month). Details on variables assigned to each category are listed in Appendix A.

The first submodel, (1) initial abundance, estimated copepodid abundance during the first sampling event. Since our initial sampling occasion occurred in August, we tested the following August values for their effect on initial abundance: trap depth, temperature at trap depth, surface light proportion at trap depth, trap position above or below the thermocline, and reservoir outflow. Not all traps were set in August due to additional trap lines and depths added during later months, which prevented us from using zooplankton covariates in the initial abundance submodel of the full dataset. Therefore, using a subset of data, we tested captures of total zooplankton and broad taxa level zooplankton abundances as covariates for August initial abundance, but we did not find evidence for these zooplankton covariates influencing S. californiensis initial abundance.

The second submodel, (2) population growth, addressed changes in copepodid abundance between months. We investigated the influence of temperature at trap site and average monthly reservoir outflow to assess covariate effects on changes in abundance between months. The third submodel, (3) detection, estimated the probability of detecting copepodids if they are present at the site. To understand the ability of the light trap method to detect a present copepodid, water clarity (via light extinction coefficient), surface light proportion at trap site, percent moon fullness, trap position above or below the thermocline, the size of the trap funnel (i.e., 4.5 mm and 5 mm holes), copepod removal during the first monthly observation, temperature at trap depth, number of sculpin captured in trap, and other zooplankton abundance in trap were all tested as covariates. We used the total abundance of non-target zooplankton as a covariate to test whether the sample size affected our ability to detect copepodids, since it could be more difficult to locate and identify copepodids while searching samples with greater non-target zooplankton abundance. We also used abundance of other zooplankton taxa as covariates, including Leptodora, Bosmina, and Daphnia spp. along with broader groups such as calanoid and cyclopoid copepods, to test whether specific groups of zooplankton affected copepodid detection. We were particularly interested in determining whether presence of Leptodora in the trap could reduce the probability of detecting copepodids since Leptodora are predatory and may consume copepodids. In addition, since copepods were removed during trapping, we tested whether our removal of copepodids during the first trapping event impacted our ability to detect them in the second trapping pass each month. We created a binary covariate denoting whether copepodids were removed from the site during the first trapping pass and used this as an observation-level covariate to evaluate the impact of our sampling. Other light trap variations, such as a trap malfunction, were not tested as there were very few instances of their occurrence (e.g., the LED light went out in less than two trap samples per month).

We fit N-mixture abundance models using the function pcountOpen in the package “unmarked” version 1.0.1 (Fiske & Chandler, 2011) in R studio statistical environment version 1.4.1103 (R Core Team, 2020). We used a negative binomial distribution, as variance in our count data was greater than our mean, and we set model dynamics to “trend,” which assumes exponential population growth from season to season. The highest single count observation for one site was 169 copepodids (during the previous test sampling year), so we set K = 300. K defines the upper bound of discrete integration. A K value of 300 was larger than the highest expected site count, and trials showed that it was a good compromise between stable model results and computational time.

Salmincola copepodid count data, environmental measurements, zooplankton community data, and trap variation data were compiled and formatted for use as covariates in the models. All continuous covariates were z-standardized so that relative effects of each covariate could be compared in model outputs. Covariates were first tested for significance individually, then combined in a stepwise fashion. Fit of these models was assessed using Akaike’s Information Criterion (AIC; Akaike, 1973) scores and individual covariate confidence intervals (95%). We considered variables significant when the 95% CI range for their effect did not contain zero. We performed a bootstrap simulation on the global model for the N-mixture abundance model to assess whether there was presence of overdispersion. We used the “Nmix.gof.test” function available from the “AICcmodavg” package version 2.3.1 (Mazerolle, 2020) and ran 1,000 iterations.

Results

We collected 675 samples from 106 sites during 2020. Of those samples, 136 contained copepodids with a total of 420 S. californiensis copepodids captured across all samples. Cougar Reservoir had the greatest number of copepodids captured in a single month (139 copepodids caught in October), while copepodid captures were relatively low in Lookout Point Reservoir, with no copepodids caught in Lookout Point Reservoir in November (Table 1). The largest number of copepodids found in a single light trap (n = 27) occurred in November at Cougar Reservoir. The most abundant non-target zooplankton taxa were calanoid copepods and Daphnia spp. Other non-target taxa in descending order of abundance were cyclopoid copepods, Bosmina, Acari (mites), Ostracod, Leptodora spp., Diacyclops thomasi, Leptodiaptomus spp., and Chironomid spp. Less frequently encountered taxa included mayflies, stoneflies, Epischura nevadensis, Holopedium spp., Scapholeberis spp., Crangonyx spp., Keratella cochlearis, and Hydropsychidae.

Initial abundance

Our top abundance model (Table 2, Model 1, and Table 3) indicated initial copepodid abundance to be influenced by initial (August) temperature and by which reservoir the copepodids inhabited. There was a negative quadratic relationship between copepodid initial abundance and temperature, with abundance increasing as water temperature-at-depth approached the mean August temperature of 15.2 °C, and then decreasing as water temperature continued to increase (Fig. 3). Reservoir was the covariate with the strongest effect on copepodid abundance (Table 3) for initial August abundance. Cougar Reservoir had the highest August abundance, followed by Fall Creek Reservoir, then Lookout Point Reservoir (Fig. 3). Lookout Point Reservoir initial abundance was significantly lower than both Cougar and Fall Creek reservoirs. The 95% confidence interval for Fall Creek Reservoir overlapped with Cougar Reservoir, indicating uncertainty regarding whether copepodid initial abundance in Fall Creek Reservoir was less or greater than Cougar Reservoir initial abundance, possibly because we were only able to sample Fall Creek Reservoir for two of the five months. At the mean August temperature of 15.2 °C, initial abundance in Cougar Reservoir was estimated at 58.6 copepodids per site, 43.8 copepodids per site in Fall Creek Reservoir, and 3.5 copepodids per site in Lookout Point Reservoir.

Table 2 Abundance models ordered by Akaike’s Information Criterion (AIC) score, with the top model having the smallest score (ΔAIC = 0).

The ‘ID’ column is the model identification number, ‘Par.’ is the number of parameters in the model, ‘ΔAIC’ is the difference between a model’s AIC score and the top model’s AIC score, and ‘AIC weight’ is the model AIC weight. Models < 5 ΔAIC are shown. The ‘Model Name’ lists covariates included in each model, with ‘IA’ refering to the initial abundance submodel, ‘G’ to the population growth submodel, and ‘P’ to the detection submodel.

ID	Model name	Par.	AIC score	ΔAIC	AIC weight	
1	IA (reservoir + initial temperature2)
G (temperature + outflow)
P (moon + thermocline + temperature + fish count)	12	1120.78	0.00	0.54	
2	IA (reservoir + initial temperature2)
G (outflow)
P (moon + thermocline + temperature + fish count)	11	1122.34	1.56	0.25	
3	IA (reservoir + initial temperature2)
G (temperature2+ outflow)
P (moon + thermocline + temperature + fish count)	12	1122.90	2.10	0.19	

Table 3 Output from the top N-mixture abundance model.

Parameters for the three submodels (initial abundance, population growth rate, and detection) are listed in separate sections. Cougar Reservoir was used as the reference reservoir for initial abundance. Data were standardized prior to model fit to make parameter estimates comparable.

Parameter	Estimate	SE	Lower 95% CI	Upper 95% CI	p-value	
INITIAL ABUNDANCE (IA)	
(Intercept)	4.07	0.29	3.50	4.64	<0.05	
Reservoir: Fall Creek	−0.29	0.26	−0.80	0.22	0.26	
Reservoir: Lookout Point	−2.83	0.45	−3.71	−1.95	<0.05	
Temperature2	−0.61	0.09	−0.79	−0.43	<0.05	
POPULATION GROWTH RATE (G)	
(Intercept)	−0.15	0.12	−0.39	0.09	0.23	
Temperature	0.18	0.09	0.004	0.36	0.06	
Outflow	−0.48	0.13	−0.74	−0.23	<0.05	
DETECTION (P)	
(Intercept)	−3.12	0.17	−3.45	−2.78	<0.05	
Moon Fullness (%)	0.25	0.09	0.07	0.43	<0.05	
Below Thermocline	−1.02	0.23	−1.47	−0.57	<0.05	
Temperature	0.78	0.13	0.53	1.03	<0.05	
Number of Fish	0.23	0.09	0.05	0.41	<0.05	
Notes.

SE Standard Error

CI Confidence Interval

Note temperature is squared for the initial abundance model.

Figure 3 Influence of temperature and reservoir on our top abundance model.

(Table 2, row 1, and Table 3) Graphs depict a negative quadratic relationship between copepodid initial abundance and temperature, with abundance increasing as water temperature-at-depth approaches 15.2 °C and then decreasing at higher temperatures. The thick solid line shows estimated copepodid initial abundance in August and the shaded area bounded by thin lines represents the 95% confidence interval for each reservoir.

Population growth

The population growth rate submodel was influenced by both water temperature at the trap site and mean monthly reservoir outflow (Tables 2 and 3). Temperature had a positive significant effect (95% CI does not contain zero) and outflow had a negative significant effect on population growth (Table 3). This means as temperature at the trap site increased, copepodid populations grew more quickly, and as reservoir outflow increased, copepodid populations grew less quickly (Fig. 4). The effect of outflow was stronger than the effect of temperature, indicating outflow has a greater influence on the population growth rate (Table 3). While holding the other covariates at their mean values, the population growth rate is estimated to be 1, indicating a stable population, at a temperature of 16.2 °C, near the middle of the overall temperature range observed (Fig. 4). However, when considering outflow, a stable population occurs at an outflow of 28.1 m3/s, in the lower range of observed reservoir discharge.

Figure 4 Model results for population growth and detection probability compared to temperature, outflow, moon fullness, position of trap relative to the thermocline, and the number of fish (sculpins) captured during light trapping.

Red dashed lines on population growth sub-models indicate the threshold of population stability (values below are declines and values above are growth). Error bars and grey shaded areas represent 95% confidence intervals.

Detection

As for the detection submodel, percent moon fullness, temperature at trap, and number of sculpins in trap were all positively correlated with detection probability while trap position below the thermocline was negatively correlated with detection probability (Table 3). Non-target zooplankton covariates, including total zooplankton abundance in trap and presence of Leptodora species, did not prove to be important covariates, determined by AIC scores and 95% confidence intervals around estimates. The mean detection probability (all covariates set to their mean value) for our sampling method was 0.042, indicating a 4.2% chance of trapping an individual copepodid if one is present at the site. If the trap was to be set under ideal controllable conditions (above the thermocline during a full moon at a warm temperature of 23.5 °C), the probability of detecting a copepodid when it is present with this method could be as high as 0.29. Fish caught in the light traps were sculpin larvae, with a maximum of 35 individuals in a single trap. Sculpin larvae abundance slightly increased detection probability. For example, the presence of five individuals in the trap increased the detection probability to 0.06 (when holding the other covariates at their means), while the largest number of sculpins in a single trap increased detection to 0.445, although this estimate had a wide 95% confidence interval (Fig. 4). Further, the bootstrap value of ĉ (ratio of observed/expected for 1,000 iterations of the global model) was 1.116 (p-value = 0.175), indicating that there was no evidence of overdispersion.

Discussion

Abundance of S. californiensis copepodids is strongly associated with temperature. Our model shows that the highest initial abundance of copepodids occurs near 15 °C. This aligns with previous observations and estimates of higher numbers of copepodids captured in traps around 16.5 °C (Murphy et al., 2023). Free-swimming copepodids have a strong negative relationship between survival and temperature, with decreased survival time at warmer temperatures (Murphy, Gerth & Arismendi, 2020). While they survive longer at cool temperatures, they may also be less active. Fifteen degrees Celsius is also within the thermal range of S. californiensis’ host, Chinook Salmon (Richter & Kolmes, 2005). Thus, the quadratic relationship we observed between copepodid initial abundance and temperature may be the result of reduced copepodid survival in warmer waters, in addition to them being less likely to encounter a suitable host at cold temperatures, as reduced swimming activity would make copepodids less likely to locate a host. Alternatively, copepodids may seek out moderate temperatures if their host species frequent these waters, since shadows and shock waves from swimming fish can elicit copepodid swimming bursts (Poulin, Curtis & Rau, 1990).

Cougar Reservoir had the largest number of copepodids captured and was also the reservoir with the greatest estimated initial abundance. Since the covariate for Lookout Point Reservoir shows significant differences from Cougar Reservoir and a stronger effect than the temperature covariate, the difference between reservoirs cannot be explained by temperature alone. Non-target zooplankton and the other covariates we tested also did not explain the differences in copepodid abundance, suggesting differences among reservoirs could not be fully accounted for by any of the environmental or biological covariates we measured. Although all three reservoirs contain suitable host species, they may occur in different densities which could affect copepodid abundances (Hasegawa & Koizumi, 2021). We did not consider the entry timings or densities of juvenile salmon, nor the reservoir-specific densities of Rainbow and Cutthroat Trout that could also support S. californiensis populations. Each reservoir also experiences different strengths in thermal stratification patterns which likely interacts with prey to alter predicted fish use, the resulting sizes of the host fishes, and the likelihood of attachment to a host, in addition to influencing the duration of the infectious stage. We were unable to estimate host fish densities and sizes concurrently with our sampling to test these hypotheses.

We demonstrate that our trapping method is an effective tool for monitoring S. californiensis populations in reservoirs. While this trapping method had been presented in a previous publication (Murphy et al., 2022), its efficacy had not yet been assessed. Since the covariates related to the microscopy portion of the detection process were not shown to be important in our models (i.e., high zooplankton abundance in the sample did not appear to be obscuring copepodids under a microscope), our “detection probability” from our top model can be a surrogate for “light trap efficacy,” since it is the probability of capturing an individual copepodid at the site. The mean detection probability for this light trap method (a trap set in mean reservoir conditions of 15.2 °C water temperature, a moon that is 51.3% full, and set above the thermocline) is 0.04, meaning there is a 4% chance of catching a single copepodid present at the site, and a 35% probability of capturing at least one copepodid if 10 are present. However, efficacy dramatically improves for traps set under more ideal conditions. For example, in a 48-hr trapping scenario, for traps set during a full moon at a depth where the temperature is 23.5 °C, detection probability may be as high as 0.29, suggesting a relatively high chance of detecting copepodids at the site if they are present, particularly if multiple traps are set or multiple copepodids are located at trap sites.

The positive relationship between moon fullness and detection probability was unexpected. While we hypothesized that more ambient light would decrease contrast between the light trap and surrounding water and thus lower detection probability, we show evidence of the opposite, as a fuller moon increased detection probability. Poulin, Curtis & Rau (1990) may offer the best explanation for this. In their study on S. edwardsii, a parasitic copepod closely related to S. californiensis, copepodids responded to shadows passing overhead (intended to simulate a nearby host fish) with increased swim speed and distance traveled. It is possible that under high ambient light conditions, our light traps cast a shadow simulating a passing host fish, thus encouraging copepodids at the site to enter the trap. Interestingly, Poulin, Curtis & Rau (1990) found no relationship between shadow intensity (defined in their study as shadow width and passing frequency) and the strength of copepodid response. The increased detection probability of our traps when ambient light is highest suggests that there is a relationship between strength of copepodid response and shadow intensity, when intensity is instead defined as shadow darkness or contrast. This could be confirmed in future studies under controlled laboratory settings.

It was also surprising that more sculpin larvae are associated with an increase in detection probability. We expected that small fishes and large invertebrates could be consumers of copepodids. However, dissected sculpin stomachs showed no evidence of consumption of copepodids (Murphy et al., 2025). This indicates a non-trophic driver of their co-occurrence. Ultimately, the samples containing sculpin larvae are small and only occur in Lookout Point Reservoir. The positive association could be because of the reasons mentioned for ambient light—these positively phototactic fish swimming toward the light traps may cast additional shadows and cause vibrations, triggering a copepodid’s host-seeking response (Poulin, Curtis & Rau, 1990). Alternatively, both sculpins and copepodids may be attracted to similar cues when they co-occur. A larger sample size, or a controlled experiment testing the effects of including fish inside the light traps, would be needed to explore these potential relationships.

Other covariate effects on detection probability are more easily interpreted. Traps located below the thermocline have a lower detection probability than those located above the thermocline. This may be due to the difference in water density on either stratum of the thermocline effectively creating a semi-permeable barrier. Although water is more stable below the thermocline due to reduced mixing from wind, it could be possible that some amount of mixing aids copepodids in locating the traps. Warmer environments are also correlated with increased detection of copepodids. Warmer water decreases the lifespan of individuals during this host-seeking copepodid life stage but increases the developmental rate for eggs and may allow for greater activity and swimming speeds (lower water density in warmer conditions). More active copepodids would be more likely to encounter the light trap and locate the funnel entrance.

Reservoir outflow and site temperature have significant influences on copepodid population growth. Outflow is associated with a decrease in population growth rate; since sampling occurred near the dam of each reservoir, this may be due to copepodids being flushed out of the reservoir. As water temperature increased at the trap site, the population growth rate also increased. At least some developmental phases of S. californiensis are temperature dependent (Murphy, Gerth & Arismendi, 2020), so it is likely that increases in temperature shortens life cycle duration, thus increasing the number of life cycles that can be completed (and copepodids produced) between months. Warmer environments may also correspond to summer and fall months with increased Chinook Salmon presence crossings near 16 °C (Murphy et al., 2023). Months with greater potential for contact with Chinook Salmon could lead to an increase in S. californiensis growth rates due to a larger number of copepodids encountering suitable hosts.

Implications and future directions

Our findings provide insights into the efficacy of this trapping method and indirect reservoir water management as a potential tool to minimize infection rates without handling threatened fish species. Many dam operators can selectively manipulate outflow and temperature of a reservoir. Our model results suggest that even moderate outflows can decrease copepodid population growth rates. By strategically increasing reservoir outflows from the temperature strata of a reservoir that correlate with highest copepodid abundance, especially during months of peak abundance, exposure of juvenile Chinook Salmon to this infectious stage of S. californiensis may be decreased.

While it is possible that flushing copepodids from a reservoir may shift infection risk from the reservoir to the downstream environment, the natural prevalence and intensity of the parasite in stream reaches would suggest this is not likely when downstream flows are similar to natural flow conditions (Monzyk, Friesen & Romer, 2015). As S. californiensis is endemic to the Pacific Northwest (Piasecki et al., 2004), running streams are the native, historical setting for the host-parasite relationship between salmonids and S. californiensis, and one would expect co-existence. We expect flow to decrease attachment, and there may be increased predation on copepodids released from reservoirs by stream-dwelling planktivorous macroinvertebrates such as Hydra sp. and net-spinning caddisflies. However, monitoring may be important to determine whether releases of copepodid-rich waters upstream increase copepodid populations in other downstream reaches or reservoirs (e.g., releases from Lookout Point Reservoir enter Dexter Reservoir directly downstream). It may be that affected reservoirs benefit from selectively releasing water to mitigate copepodid populations concurrently or in sequence. Our sampling method may be useful in monitoring such situations, as well as for implementing new copepodid monitoring programs and assessing the effect of management actions in other stratified reservoirs.

Our sample size is limited in two ways: (1) by the time and logistics of setting or retrieving ∼90 traps in a single day in three large reservoirs that are separated by hours of road travel, and (2) by the amount of time it took to manually search each sample for copepodids. Using alternative methods of sample processing for future trap deployments, such as genetic (e.g., environmental DNA) or automated morphological methods, to detect and quantify S. californiensis could allow for more sampling with reduced processing times, thus increasing sample size and potentially improving model fit while reducing costs.

Calibrated genetic tools may reduce the time and expertise required to morphologically identify S. californiensis. Quantitative polymerase chain reaction (qPCR) is one such genetic method that could be faster and more cost effective since it allows for the simultaneous amplification and quantification of DNA sequences specific to a target species. A recent study reported a qPCR assay was established for S. edwardsii and validated in the field using electrofishing methods (Katz et al., 2023). However, while nucleotide sequences for S. californiensis are readily available in GenBank (Ruiz et al., 2017), it is unknown whether the currently available sequences would be able to differentiate between this species and others. For example, a historically co-occurring species of gill maggot on Mountain Whitefish (Prosopium williamsoni) recently discovered in archived fish collections from the Willamette Valley has not yet been genotyped (Murphy et al., 2020a). Additionally, our light trap method collects very high volumes of non-target zooplankton. Even after filtering samples by size with mesh sieves, a large amount of non-target genetic material would remain to be processed by qPCR. Other collection methods, such as Van Dorn sampling and plankton tow net sampling, failed to collect any S. californiensis specimens in previous sampling and are even more biased towards non-target taxa (Murphy et al., 2022). Capture or processing methods may need to be modified to reduce the number of non-target zooplankton while retaining the relatively rare S. californiensis copepodids for qPCR methods to be successful.

Another method that could be adapted for faster sample processing is an automated morphological method known as automated flow imaging microscopy, which uses laser detection to capture digital images of organisms in fluid, then analyzes those images to identify the organisms and estimate abundances (Stanislawczyk, Johansson & MacIsaac, 2018). While this method can detect and identify to genus level relatively rare organisms within our target size range, this method relies heavily on well-developed libraries of reference images and that species are sufficiently morphologically distinct as automatic classification systems are still error prone (Xiong et al., 2020). To our knowledge, these methods have not been applied to freshwater parasitic copepods, so library development and quality control by a traditional identification method would likely be needed to reliably employ automated identification.

Conclusions

The goal of this research was to develop predictive models of S. californiensis copepodid abundance in relation to various environmental factors, as well as evaluate light trapping as a sampling method used to collect the copepodids. Both goals were achieved using N-mixture abundance modeling. Our models showed that the most important predictor of copepodid abundance measured during sampling was site temperature. Copepodid initial abundance increased as temperatures approached 15.2 °C, and then decreased as temperatures continued to warm. We also found that warmer temperatures increased copepodid population growth rate from month to month, while higher reservoir outflow decreased population growth rate. These are significant findings, as temperature and outflow are factors that may be manipulated in many reservoirs by dam operators. Our models were also able to estimate the efficacy of our sampling method and the factors that influence that efficacy. When set across a wide range of conditions similar to our sampling layout, the mean probability of this method to detect a single copepodid at a site when present was low (4%). However, sampling optimized for maximum efficacy, i.e., setting a trap during a 48-hour period that aligned with the full moon and at a depth where the temperature was near 23.5 C, increased detection to 29%. To our knowledge, this is the only sampling method with a known efficacy that can consistently detect the free-swimming copepodid of S. californiensis. Thus, this method may be an important tool for monitoring the presence and abundance of this salmonid parasite in reservoirs and potentially evaluating future management actions.

Supplemental Information

Supplemental Information 1 Covariate types, definitions, and descriptions of covariates tested along with photos of a light trap

Photos of a light trap depicting (counterclockwise from top left) the funnel and trap entrance, the LED light used to attract Salmincola californiensis copepodids, and the outside body of the trap with rope harness for attaching to an anchored line. Photos C.A. Murphy.

We thank Bill Gerth, Jennifer Boland and other members of Arismendi laboratory at Oregon State University for their invaluable help in zooplankton identification and ecological input. We also thank Greg Taylor and Todd Pierce of the US Army Corps of Engineers for aiding our access to sample sites and Dave Leer for his help and ingenuity in the field. Any use of trade, firm, or product names is for descriptive purposes only and does not imply endorsement by the US Government.

Additional Information and Declarations

Competing Interests

Author Contributions

Field Study Permissions

Data Availability

The authors declare there are no competing interests.

Kelsi Antonelli performed the experiments, analyzed the data, prepared figures and/or tables, authored or reviewed drafts of the article, and approved the final draft.

Christina Murphy conceived and designed the experiments, performed the experiments, analyzed the data, prepared figures and/or tables, authored or reviewed drafts of the article, and approved the final draft.

Amanda Pollock performed the experiments, analyzed the data, prepared figures and/or tables, authored or reviewed drafts of the article, and approved the final draft.

Ivan Arismendi conceived and designed the experiments, analyzed the data, prepared figures and/or tables, authored or reviewed drafts of the article, and approved the final draft.

The following information was supplied relating to field study approvals (i.e., approving body and any reference numbers):

Fish were collected under Oregon State University IACUC # 5163 and NOAA and ODFW permit #22982.

The following information was supplied regarding data availability:

The dataset is available at ScholarsArchive@OSU: Antonelli, K., Murphy, C.A., Pollock, A.M.M., and Arismendi, I. (2024). Salmincola californiensis light trap dataset in Willamette Reservoirs, 2020 (Version 2) [Data set]. Oregon State University. https://doi.org/10.7267/sb397h833.

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
