# Peer review of "Environmental conditions influencing the abundance of the salmonid ectoparasite Salmincola californiensis across upper Willamette River Reservoirs, Oregon"

_PeerJ, doi:10.7717/peerj.19228_

## Round 0.1 · original submission · Major Revisions

Your manuscript focuses on a topic of interest to PeerJ. However, some sentences and paragraphs have been clarified to reinforce the ideas to be conveyed or to avoid ambiguity. I encourage you to revise the article taking into account all the improvements suggested by the reviewers.

Reviewer 1 ·

Basic reporting

Dear authors,
This manuscript examines the environmental factors potentially affect the copepodid abundance of Salmincola californiensis. They also report the light trap efficiency. As authors mentioned, this parasite is harmful to endangered salmonids, and their findings are very important for managements and control measures. Their field study is well designed, and statistical analyses look reasonable. The structure of this manuscript is well constructed, and most of the results are discussed properly (but see comments below). I hope my comments will be useful for strengthening the manuscript, and it can be accepted after revisions.


Major comments

L 116
It is unclear how the authors expect high ambient light from the surface and a bright moon reduce lower detection probability? The discussion clarifies this point, however, the authors should note the rationale behind this prediction here.

L265, 274
Can the authors more clearly explain each sub-model structure by using sub-headings? e.g., (1) initial abundance, (2) population growth…. The current structure is somewhat difficult to follow. For the results of each model (L 308), I suggest using similar sub-headings.

L308
Can the authors provide a brief summary of how many species of plankton (and their proportions in the samples) were captured in this trap during the study period? I understand this is not a central result, but this would be valuable information for readers.


Minor comments

L 32
Remove “the” (overlapping).

L67
Not only fin rays, but also fin bases or fin membrane can be their attachment sites (so just mention “fins”)

L 95
The below paper also reported the environmental factors potentially affecting Salmincola sp., and hence authors could cite and discuss based on their findings?

Hasegawa, R., & Koizumi, I. (2021). Relative importance of host‐dependent versus physical environmental characteristics affecting the distribution of an ectoparasitic copepod infecting the mouth cavity of stream salmonid. Ecological Research, 36(6), 1015-1027.

L540
Authors are recommended to cite original article by the same author, not the thesis.
Herron, C. L., Kent, M. L., & Schreck, C. B. (2018). Swimming endurance in juvenile Chinook Salmon infected with Salmincola californiensis. Journal of Aquatic Animal Health, 30(1), 81-89.

Experimental design

Major comments

L 167
I’m not insisting on this comment, but I suggest the authors provide a figure (photo) of the light trap. It is difficult to imagine the trap's structure without reading previous paper (Murphy et al., 2022).

L 270
In the Introductions, authors predicted that only Leptodora spp. can affect the recorded copepodid abundance because this group potentially preys on copepodids. However, authors used the total abundance of all non-target zooplankton in their models. Can the authors construct another model using the abundance of Leptodora spp. only? Or are there any reasons for using the non-target zooplankton abundance (Perhaps, authors aimed to test the effects of non-target zooplankton abundance on the detectability of Salmincola copepodids? L 368). If so, please clarify why the total abundance of non-target zooplankton was used.


Minor comments

L139
Conservation for what?

L162
I suggest “the N-mixture abundance modeling approach (see below; Royle, 2004)”.

L186
Provide how much percentage of the ethanol used (70% or 100%?).


L 199
Provide the methods for measuring these environmental factors (water temperature, light, depths).

L 206
Provide the unit of outflow (probably, m3/s?)

L 289
Provide the package version numbers.

L 314
I suggest providing “model ID” in Table 2 (i.e. model 1, model 2…), so that authors can cite model ID in this sentence.

Validity of the findings

Major comments

L 308
According to their results, infection levels were heterogeneous among reservoirs, and especially, the Cougar reservoir originally had higher infection levels than those of others. Can authors explain this result? Does each reservoir have different environmental characteristics or host density?

L316
All reservoirs show the very similar patterns (Figure 3: infection peak around 15.2), which is interesting. However, authors did not discuss the results specifically. In particular, can the authors explain why copepodid abundance increased toward 15.2℃ but decreased after 15.2℃? I think authors can (roughly) explain these results based on previous papers described the life cycle of Salmincola.


Minor comments

Table 2
I recommend presenting only the top three models because the delta AIC increased substantially after the third-ranked model. Moreover, model names “LAM” and “GAM” should be changed to other names (e.g. IAM and PGM, respectively). Otherwise, readers would misunderstand those as “linear model” or “generalized additive model”.

L 457
Authors can provide recent citations about eDNA of Salmincola, such as “Katz, A. D., Tetzlaff, S. J., Johnson, M. D., Noble, J. D., Rood, S., Maki, D., & Sperry, J. H. (2023). Molecular identification and environmental DNA detection of gill lice ectoparasites associated with Brook Trout declines. Trans. Am. Fish. Soc. 152(6), 788-808.”

Figure 1
Because this figure looks very similar to the figure provided by Kabata & Cousens (1973), I don’t think authors have to provide it (I recommend just citing Kabata & Cousens 1973). If authors want to provide this new figure in this manuscript, they should add more relevant information such as each developmental period for each stage.

Figure 4
Remove # in the panels juvenile fish – Detection probability.

Reviewer 2 ·

Basic reporting

The manuscript is mostly well written, however there are some grammatical issues. Please run a spell check on the document and this will catch repeats of words (the the) and misspelling of the Latin name of your copepod (both in the abstract). Reservoir should be capitalized when used as a proper noun (Cougar Reservoir). The authors do a good job of not capitalizing reservoir when it is used to describe several of them (“Cougar, Lookout Point, and Fall Creek reservoirs”). Also, be consistent with using Reservoir- I think it is appropriate to include Reservoir each time you identify the reservoir name (i.e., "Cougar Reservoir" instead of "Cougar"); or you can state in the beginning that you will refer to them with this short name.

Abstract
Line 26. What about other body parts where the parasite attaches? Do these negatively affect the hosts, too?

Intro
Line 57 “Parasites…” an awkward jump from the previous/next sentences. Maybe you meant for this sentence to be elsewhere? Tt feels shoehorned in.
Line 58 “This” should be followed by a noun. What can shift? Unclear.
Line 82. Capitalize Basin.
Line 109. “reservoirs located at the upper WRB”. Should this be “in” the upper WRB?

Materials and Methods
Line 136. Fix punctuation at “western OR, USA”.
Line 138 “primarily for flood”. I’m pretty sure the USACE calls this “reducing flood risk” these days. Missing USACE 2022 in the ref. Are you referring to the three USACE web pages in your reference section? Those are odd reference formats. Confirm the journal style and cite appropriately in the text. They are not currently cited in the text.
Line 145. Need topic sentence.
Line 147 remove hyphen
Line 153 list which river for FC?
Line 190 please clarify a total of 30 m apart maximum
201 We recorded light trap characteristics…

Discussion
359 indent first line
364 “is an effective”
390 increased
393 laboratory
420 increases (change both occurrences to past tense?)

Conclusions
500 and 502 use the EM dash (the longest one) when a break in the text (instead of a hyphen).

References
There are several references with a different year in the in-text citation compared to the references section. Does this journal allow DOI numbers for each reference? If so, please add. Several references listed are not cited in the text. In-text citations are inconsistent with punctuation.
Missing USACE 2022.
In-text citation Maserolle, 2020, but ref section has 2019.
In-text citation Xiong, 2019, but ref section has 2020
Several references include “et al.” List these coauthors authors?
Stanislawczyk not cited in text
Kabata, Z. & Cousens, B. not cited in text
Fasten not cited in text

Figures
Fig 3 caption is cut off.

Experimental design

no comment

Validity of the findings

no comment

Additional comments

In general, the manuscript was well written, and the research well executed. I think these findings will help guide USACE in reservoir management and the monitoring of these parasites. There are some minor edits, mostly grammatical, that need to be addressed prior to publication.

---

## Round 0.2 · Minor Revisions

Dear authors,

Thank you very much for improving your manuscript.

However, there are still some minor corrections that should be made to improve its quality.

Best regards,

Reviewer 1 ·

Basic reporting

Dear authors,

Many thanks for your attention to my previous comments. The authors have addressed the comments nicely. At this stage, I only have several minor concerns.
Because the line numbers are different between a PDF and a Word file including marked changes, I used the numbers in PDF.


Minor concerns

L 20
Remove “trout”

L 27
lifestage → life stage

L 55
Can the authors add some citations for this sentence?

L 65
Can the author briefly explain on males (dwarf forms)?

L 87
Number of parasites per host → Number of parasites per infected host
(Please check the definition of intensity defined by Bush et al. 1997)

L 271
Can the author say “three submodels using the following three response variables”?
I’m not familiar with this N-mixture model, but possibly, these three variables (1-3) were used as response variables in each model, so I think authors can clarify this point.


L 327&367
Although authors noted that variables did not overlap with zero considered as significant effects (L 367), they should note this in Methods (last paragraph in Methods, for instance).

L 340
“spp.” should not be Italic, revise through the manuscript.

L 528
This sentence is irrelevant to the topic. While the previous sentence discusses if the currently available sequences are able to differentiate between one species and another species, this sentence discuss the deficiency of the currently available sequences. Remove this sentence.

Table 2
Maybe, “Akaike’s information criterion” is correct. Please check if this expression is correct.

Table 2
At the first round, I suggested presenting only the top three models because the delta AIC increased substantially after the third-ranked model. The authors replied as they corrected it, but Table 2 in the current manuscript is obviously the same as the previous version. Please check and revise it (if the authors would like to keep the current version showing the top 10 models, I respect their decision).

Experimental design

No comments

Validity of the findings

No comments

Additional comments

No comments

Reviewer 2 ·

Basic reporting

The manuscript is greatly improved after the first round of peer review.

Experimental design

The manuscript is greatly improved after the first round of peer review.

Validity of the findings

The manuscript is greatly improved after the first round of peer review.

---

## Round 0.3 · accepted · Accept

The authors have thoroughly addressed all of the reviewers' comments, incorporating the necessary revisions to improve the manuscript. I have personally evaluated the changes and find them satisfactory. The manuscript now meets the required standards, and I am confident that it is ready for publication.